# The Evolution of Minimally Invasive Spine Tumor Resection and Stabilization: From K-Wires to Navigated One-Step Screws

**DOI:** 10.3390/jcm12020536

**Published:** 2023-01-09

**Authors:** Mai Shiber, Gil Kimchi, Nachshon Knoller, Ran Harel

**Affiliations:** 1Department of Neurological Surgery, Sheba Medical Center, Ramat Gan 52621, Israel; 2Sackler Medical School, Tel-Aviv University, Tel Aviv 69978, Israel; 3Adelson School of Medicine, Ariel University, Ariel 40700, Israel

**Keywords:** spine tumors, minimally invasive surgery, navigation guidance, one-step screws, running head, the evolution of minimally invasive spine tumor resection

## Abstract

Minimization of the surgical approaches to spinal extradural metastases resection and stabilization was advocated by the 2012 Oncological Guidelines for Spinal Metastases Management. Minimally invasive approaches to spine oncology surgery (MISS) are continually advancing. This paper will describe the evolution of minimally invasive surgical techniques for the resection of metastatic spinal lesions and stabilization in a single institute. A retrospective analysis of patients who underwent minimally invasive extradural spinal metastases resection during the years 2013–2019 by a single surgeon was performed. Medical records, imaging studies, operative reports, rates of screw misplacement, operative time and estimated blood loss were reviewed. Detailed description of the surgical technique is provided. Of 138 patients operated for extradural spinal tumors during the study years, 19 patients were treated in a minimally invasive approach and met the inclusion criteria for this study. The mortality rate was significantly improved over the years with accordance of improve selection criteria to better prognosis patients. The surgical technique has evolved over the study years from fluoroscopy to intraoperative 3D imaging and navigation guidance and from k-wire screw insertion technique to one-step screws. Minimally invasive spinal tumor surgery is an evolving technique. The adoption of assistive devices such as intraoperative 3D imaging and one-step screw insertion systems was safe and efficient. Oncologic patients may particularly benefit from the minimization of surgical decompression and fusion in light of the frailty of this population and the mitigated postoperative outcomes associated with MIS oncological procedures.

## 1. Introduction

Over the past decades, earlier diagnosis of cancer and improved treatment modalities have resulted in a continuing rise in life expectancy of oncologic patients [1,2]. Consequently, the prevalence of spine metastases is increasing. Due to the frailty of this patient population and its increased risk for post-operative complications, the 2012 Oncological Guidelines for Spinal Metastases Management [3] have advocated the use of minimally invasive surgery (MIS) techniques for the treatment of spinal metastases. MIS techniques were associated with reduced blood loss, lower infection rates, reduced overall complication rates, improved post-operative pain control, faster recovery, and shorter length of hospital stays [4,5,6]. Our group has previously published our experience with MIS decompression and fusion for spinal metastases, assisted by fluoroscopy and K-wire guided screws [7]. Since then, the uses of intra-operative imaging and navigation devices have become prevalent in a plethora of surgical techniques, allowing for a more efficient surgical course. Recently, one-step screw systems were introduced, allowing for a simpler and faster percutaneous screw introduction.

This paper will describe the evolution of techniques for the excision of spinal metastases and instrumentation that were implemented in our department.

## 2. Methods

A retrospective analysis of 19 patients who underwent minimally invasive extradural spinal tumors resection at a single institution by the senior author (RH) was performed. Patients operated in an open approach were excluded (*n* = 119). Patients were operated using the described MIS technique if they had a tumor confined to the vertebral body and pedicle, with or without canal involvement. Only focal disease confined to one or two vertebral bodies was included and if the surgeon considered unilateral approach adequate for tumor resection. Tumors involving the canal bilaterally, the lamina, spinous process, or posterior soft tissue were considered for open surgery. All cases were presented for discussion in a multi-disciplinary tumor board, and the surgical approach was agreed upon. Following the approval of the Institutional Review Board, the authors evaluated patient records and imaging studies of patients who were operated on between November 2013 and March 2019. In addition to patient demographics, the tumor location, number of spinal levels involved, and tumor pathology were gathered. Operation-related data were collected, including estimated blood loss (EBL), duration of operation, navigation use, and screw insertion method. The primary outcomes were perioperative and postoperative complications, discharge status, duration of hospitalization, recurrence rate, and mortality. Adjuvant radiation therapy was recorded. Statistical analysis was performed with SPSS v.22 software version 22 (IBM Corp, Armonk, NY, USA). For parametric variables, data are expressed as mean and range, and for nonparametric variables, data are expressed as frequency and percentage. The univariate analysis was done with the chi-square test to assess the statistical significance (*p* < 0.05).

### Surgical Technique

The surgical technique for the minimally invasive resection of spinal extradural metastases used in the initial cases was described by the authors in 2015 [7]. To summarize briefly, the patient was positioned in the supine position, following preparation, draping and utilizing fluoroscopic guidance percutaneous K-wires were inserted to the level above and below the index vertebrae, as well as to the index vertebrae contralateral to the decompression side. A minimally invasive expandable tubular retractor was positioned under fluoroscopic guidance (X-tube, Metrix, Medtronic, MN, USA) over the facet, lamina, and transverse process on the decompression side (Figure 1A). Using a high-speed drill and a transpedicular approach, the thecal sac was exposed and decompressed, and a partial corpectomy was performed (Figure 1B, Appendix A). The retractor was then recovered to be followed by serial dilatation over the previously inserted K-wires and placement of percutaneous screws (Sextant FNS or Longitude FNS screws, Medtronic, MN, USA). Under fluoroscopic guidance, PMMA was injected through the screws to the index level and to the level above and below. Finally, percutaneous rod insertion and locking took place. The wounds were irrigated and closed after fluoroscopic verification of hardware final position.

The introduction of intraoperative 3D imaging modalities combined with intraoperative navigation allowed for modifications and improvement of the surgical technique. After positioning, the O-arm was introduced, and scanning and parking positions were determined and saved. A reference frame was attached to the iliac wing by a designated iliac pin (Medtronic, MN, USA). Following patient draping, an O-arm 3D scan was performed and transferred to the navigation station (S7, Medtronic, MN, USA). A navigated Jamshidi needle was used to insert K-wires to the levels above, below, and contralateral to the decompressed index side. A navigated dilator was used to position the X-tube retractor over the lamina and facet of the index level. Navigation-assisted drilling was performed to decompress the canal and remove extensive parts of the vertebral body. Once decompression was finalized, the retractor was removed, and navigated screws were inserted over the K-wires. A 3D scan was repeated to confirm hardware position. PMMA was injected into the instrumented levels under lateral 2D fluoroscopy using the O-arm device. The screws were connected with rods, and the rods were locked in place. Two-dimensional images confirmed the rod position. 

The emergence of one-step screws enabled further progression of the described technique. Following the initial scan, the retractor was positioned, and decompression was performed. A second scan was acquired and transferred to the navigation system. One-step screws (Viper-prime FNS, J&J, NJ, USA) were fitted with universal reference frames (Sure-trak, Medtronic, MN, USA) and calibrated accordingly (Figure 1C,D). The internal K-wire was advanced to be 3–4 mm proud relative to the screw tip. Skin incision was performed according to the navigation proposed entry point, and the screw was advanced through the muscles to the transverse process. Once navigation confirmed the screw to be in an acceptable starting point, the internal sharp K-wire was hammered into the starting point while gradually advancing the K-wire. Once docked in the starting position, the handle was used to screw the self-drilling screw into the pedicle and vertebra. A third 3D scan verified the screw positions, followed by PMMA injection and rod insertion as previously described.

## 3. Results

Overall, 19 patients were included in this study. Table 1 summarizes the patients’ demographics, neurological status, and pathological diagnoses. Nine operations (47%) took place in 2013–2014, six (31.6%)) in 2015–2016, and four (21%) in 2017–2019. Apart from one patient, all patients were treated for a single-level pathology. The most common pathological diagnosis was RCC (*n* = 5). All patients presented with spinal cord compression or nerve root impingement secondary to metastatic lesions.

The intraoperative imaging techniques included C-arm fluoroscopy (68.4%) and O-arm (31.6%) (Table 2). Sixteen patients underwent screw fixation, out of which 13 cases had short construct instrumentation using Longitude FNS screws (Medtronic, MN, USA), one (1) had percutaneous pedicle screw-rod fixation using the Sextant system (Medtronic, MN, USA), and two (2) had single-screw insertion with the Viper prime system (Johnson & Johnson, NJ, USA). The mean EBL was 368 mL (range: 0–1800). The mean operative time was 140.3 min. There were no intraoperative complications apart from one patient with excessive bleeding. The length of stay ranged between 1 and 14 days (mean 4.2 days). Eighteen patients were discharged to their homes and one to a rehabilitation facility. Recurrence was documented in six patients during a mean follow-up period of 14.3 months. The mortality rate during the follow-up period was 74% (*n* = 14). The 6-month mortality rate was 37% (*n* = 7). The postoperative complications rate was 16% (*n* = 3; respiratory infection, deep wound infection, and deep vein thrombosis). Four patients had neurological improvement following surgery, and one deteriorated neurologically. The adjuvant and neoadjuvant radiation treatments are presented in Table 2.

Figure 2 demonstrates the 6-month mortality rate according to the year groups. The differences between the groups considering the year of surgery are significant, as indicated by a *p*-value of 0.03 (calculated by the chi-squared test). Evolution of the surgical technique is demonstrated in Table 3.

## 4. Discussion

Bone involvement is the third most common site for metastatic spread, following the pulmonary and hepatic systems [8]. Their prevalence is expected to rise further with improvements in systemic control and with technological advances that allow for earlier diagnoses and increased survival of the in oncological population [9]. About 90% of cancer patients with spinal metastases report bone and/or axial back pain, accompanied with radicular pain. Half of these patients have sensory and motor deficits and more than 50% have bladder and intestinal dysfunction [10]. Treatment options for symptomatic patients with metastatic epidural spinal cord compression (MESCC) include a combination of corticosteroids [11], radiotherapy [12] or neural elements decompression through resection of compressive tumors in selected patients with or without instrumentation [1,8,13]. Although most patients may benefit from non-surgical treatment options, patients with unstable spinal column or severe spinal cord compression are likely to improve their quality of life following surgery [14].

Patchel et al. [13] demonstrated that patients with spinal metastases that have undergone surgery and radiotherapy were ambulatory for longer periods than patients treated with radiotherapy alone. Even so, unclarities still exist regarding the selection criteria for surgery. In part, this may be attributed to the general medical status of oncologic patients, in whom neoplasm-related co-morbidities such as anemia, impaired immune system and malnourishment are common, rendering open surgery to be deemed high risk [14,15]. Open spine surgery may be complicated by significant blood loss, lengthy hospital stays, high rates of post-operative infections, severe back muscle injury and the need for intensive pain control [7]. The oncological guidelines for the management of malignant extradural spinal cord compression [3] state that since surgery is associated with significant morbidity, the patients’ prognosis should be considered in the decision making process, so that patients with favorable prognosis who may be operated safely should be referred to surgery. Of note, the guidelines emphasized that every effort to minimize the surgical extent should be made in light of the advantages of the minimally invasive technique.

In recent years, technological innovation has enabled a rising number of surgeons to routinely opt for minimally invasive spinal surgeries (MISS) for the treatment of various neoplastic lesions. Using tubular retractors and microscopic visualization, MISS enables decompression of the spinal cord and nerves via small incisions, while stabilizing the spine by inserting percutaneous screws [10,16,17]. MISS techniques have been associated with reducing the risks of open spinal surgery, since they were associated with decreased intraoperative blood loss, improved wound healing and shorter postoperative hospitalization [5,6]. Therefore, MIS approach has the potential to minimize open surgery related morbidity. Consequently, minimization of the surgical approach may allow for frail patients who were not considered operable in the past to undergo surgery with in a safe and efficient manner [17]. A pivotal drawback of surgery for metastatic patients is the discontinuation of chemotherapy and radiation therapy to avoid wound dehiscence and infections [18]. MISS for spinal metastases was associated with earlier post-operative radiation therapy [19] and chemotherapy [6].

A plethora of surgical techniques were developed to achieve the primary goal of surgery, namely decompression of the neural elements and stabilization of the spinal column. Until recently, open surgical approach was the mainstay of treatment. Mini-open approach has been introduced over the past decade allowing for the tumor resection to be performed through a familiar midline approach augmented by percutaneous screws [20]. Saddeh et al. [21], compared mini-open approach to open surgery for spinal metastases and concluded that the mini-open approach was found to reduce postoperative pain and length of recovery. Minimally invasive tumor resection has been described over the last decade either with the use of tubular retractors or expandable retractors [22,23]. the introduction of spinal radiosurgery over the recent years has raised the need for separation surgery with adjuvant SRS [2]. These are well performed by MIS epidural decompression.

On 2015, Harel et al. [7] described their experience with MIS expandable tubular retractor approach instrumented with short segment fenestrated screws and PMMA augmentation. Since the description of this technique, two main technological advances were made available: intra-operative imaging and navigation, and one-step screw insertion systems. MIS surgery compromises the surgeon’s ability to comprehend the anatomical structures by direct sight, and relies significantly on either 2D fluoroscopy or 3D navigation. The 3D imaging combined with intraoperative navigation provides better visualization of the anatomy and therefore improves surgeon’s orientation during surgery and ameliorates screws localization and placement [24,25,26]. In the current series, the use of fluoroscopy was abandoned in favor of intraoperative 3D imaging and navigation. Intraoperative navigation for spine tumor resection was first described by Kalfas on 2001 [27], emphasizing the importance of surgeon’s validation of the navigation system accuracy before relying on it. Fujibayashi et al. [28] described the use of navigation for osteotomies during en-bloc tumor resection. Intra-operative navigation was utilized for osteoid osteoma curettage, enabling the surgeon direct access to the tumor, thus limiting the extent of the exposure without compromise of the spine stability [29,30]. As earlier systems relied on pre-operative CT scans and registration was cumbersome and not accurate, intra-operative navigation was rarely used. The integration of intra-operative 3D imaging allowed for intra-operative imaging with more accurate registration, and increased the popularity of this technology among surgeons. A multicenter study described the results of 50 patients with spinal tumors that had undergone open surgery with intra-operative imaging and navigation [31]. The authors concluded that tumor dissection is targeted while minimizing the dissection. In addition, utilizing intra-operative imaging and navigation for MISS tumor resection reduces the surgical team’s radiation exposure [31,32,33]. Intraoperative navigation was repeatedly shown to increase screw accuracy rate in multiple studies [26,34,35,36]. None of the patients in the current study had experienced screw mispositioning with either fluoroscopic guidance or 3D imaging guidance.

Metastatic cancer patients commonly suffer from poor bone quality, which is further exacerbated by the combination of radiation and chemotherapy, leading to lower rates of bone healing [37] and places the patients at high risk for screw pullout. Moussazadeh et al. [38] reported of 44 patients undergoing short segment instrumentation with cement augmented screws following pathological vertebral fractures, and described their practice of injecting cement to the vertebral body prior to screw placement. Differently, fenestrated screws allow cement injection through the screw after placement directly into the surrounding body, thus increasing pull-out strength while reducing the number of tools inserted into the vertebral pedicle [7,39,40]. We opted to insert fenestrated screws augmented with cement in all instrumented patients, to obtain maximal purchase in the level above and below the index vertebra. In addition, the contralateral side of the decompression in the involved vertebra was instrumented with fenestrated screw, allowing for augmentation of the residua of the vertebral body with cement to enable a three-point fixation construct. None of the constructs in this series had failed during the follow-up period.

The recent development of one-step screws allowed the mentioned technique to further advance, by simplifying the screws insertion technique and render numerous operative steps redundant [41,42]. While the standard MIS screw insertion technique involved, the insertion and removal of cannulated instruments over a K-wire, the one-step insertion screws consist of one navigated screwdriver with an integrated K-wire that corresponds to the navigation system. The integrated wire eliminates the possible complications involved with wire techniques. The principals of conversion of open surgery to MISS includes maintaining the goals and outcomes of surgery while reducing complications rate and expediting recovery. In the current study, one-step screws have been used since 2017. Screw malposition was not observed with any of the screw type and no intraoperative complications involving screw insertion were observed, thus achieving the role of MISS of obtaining the operative goals while refraining from complications.

The pivotal role of a methodological patient selection process is a key element in the treatment of oncologic patients. In light if the lower complication rated associated with minimally invasive approaches, patients in a lower functional status and a shorter life expectancy may be referred to surgery [43]. In our institution, efforts are made to implement this in the multidisciplinary decision making process. Wright el al. [44] collected data from 22 centers comparing 2001 patients that had undergone surgery for symptomatic spinal metastases between 1991–2016. The data analysis revealed that the long-term survival improved significantly over the time course of the study. The authors concluded that the change is due to earlier diagnosis, better adjuvant therapy and an improved understanding of spinal metastasis disease which results in selecting operable patients with better potential of long-term survival. This was further supported by another paper showing improved survival for patients who underwent surgery while being in a relatively good pre-operative physical condition [45]. Over the study years, the patient selection criteria have leaned towards patients with favorable prognosis. This trend may act as a two-way sword by reducing the number of patients eligible for oncological spine surgery on one hand, but significantly improving the 6-month survival rates on the other. Minimally invasive spine metastases surgery may broaden the spectrum of patients eligible for surgery compared to the open approach.

## 5. Conclusions

MISS technology is constantly evolving, allowing for improvement in surgical techniques and possibly better outcomes. MISS for spinal tumors has multiple advantages, and a wider adoption of new technology may be beneficial for these patients as a means to reduce complications and allow for faster post-operative initiation of adjuvant therapy.

## 6. Limitations

This is a retrospective study of a small cohort examining a new technology. The small numbers do not allow definite conclusions. The scope of this paper is limited to the description of the benefits of a new method. As the patient selection process for the MISS procedure is at the surgeon’s discretion, selection bias is a main limitation for this study.

## Figures and Tables

**Figure 1 jcm-12-00536-f001:**
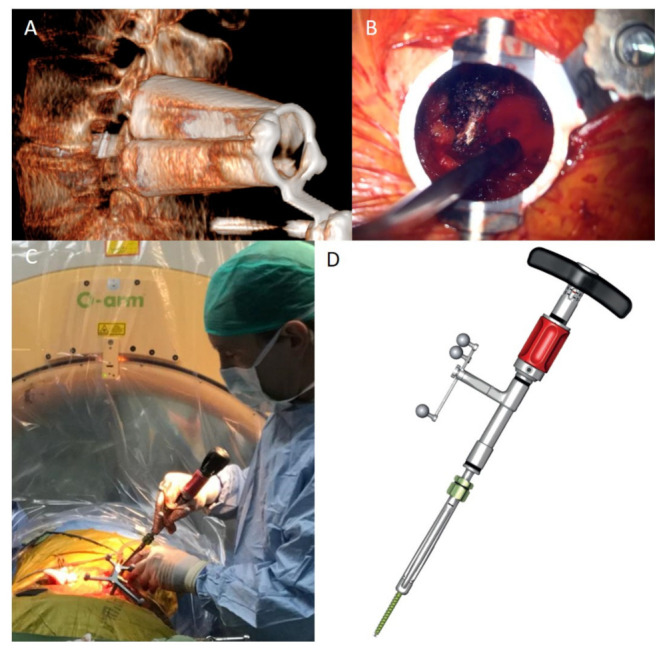
Expendable MIS retractor is positioned over the lamina and facet joint of the index vertebra allowing for a transpedicular approach to the vertebral body and partial corpectomy, thus decompressing the thecal sac and nerve roots ((**A**) 3D reconstruction of intraoperative O-arm images (**B**)); following decompression, one-step screws (Viper-Prime FNS, J&J, NJ, USA) are fitted with a Sure-trak reference frame (Medtronic, MN, USA, (**D**)) and calibrated. Following skin incision, the screws are advanced to the starting point with navigation guidance. The k-wire was advanced 2–3 mm, and a malate was used to anchor the screw. The screw handle was used to advance the screw with navigation guidance (**C**).

**Figure 2 jcm-12-00536-f002:**
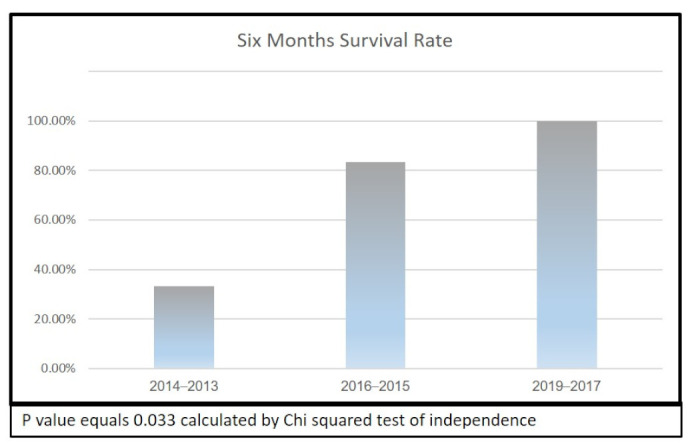
Six-month survival rate histogram for 19 patients with epidural spinal tumors operated by MISS.

**Table 1 jcm-12-00536-t001:** Demographic data from 19 patients with epidural spinal tumors.

Variable	Classification	Total
Mean age in years (range)		60 (41–82)
Sex	Female	11 (57.9%)
	Male	8 (42.1%)
ASA	2 3 4	5 (26.3%) 1 (5.3%)
ASIA	C	4 (21.1%)
	D	5 (26.3%)
	E	10 (52.6%)
Risk factors	Smoking	4 (21.1%)
	DM	7 (36.8%)
	IHD	1 (5.3%)
	HTN	8 (42.1%)
Year of surgery	2013–2014	9 (47.3%)
	2015–2016	6 (31.6%)
	2017–2019	4 (21%)
Spinal level	Cervical	1 (5.1%)
	Thoracic	5 (26.3%)
	Lumbar	13 (68.4%)
Median number of spinal levels (range)		1 (1–2%)
Pathology	RCC	5 (26.3%)
	TCC	2 (10.5%)
	NSCLC	2 (10.5%)
	Breast cancer	2 (10.5%)
	Nosapharyngeal adenocarcinoma	1 (5.3%)
	Hepatobiliary carcinoma	1 (5.3%)
	Follicular lymphoma	1 (5.3%)
	Hemangioma	1 (5.3%)
	Leiomyosarcoma	1 (5.3%)
	Carcinoma of colon	1 (5.3%)
	Phosphouretic mesenchymal tumor	1 (5.3%)
	Cholangiocarcinoma	1 (5.3%)

DM = diabetes mellitus, IHD = ischemic heart disease, HTN = hypertension, RCC =Renal cell carcinoma, TCC = Transitional cell carcinoma, and NSCLC = non-small-cell lung cancer.

**Table 2 jcm-12-00536-t002:** Perioperative data in 19 patients with epidural spinal tumors.

Variable	Classification	Total
Imaging	Xray	%
	O-arm	
Fixation Type	No fixation	3 (15.8%)
	Sextant	1 (5.3%)
	Longitude	13 (64.8%)
	Viper	2 (10.5%)
Mean EBL in mL (range)		368 (0–1800)
Mean surgery duration in min (range)		140.3 (46–221)
Anesthesia duration in min (range)		198.8 (122–279)
Intraoperative complications	Dural tear	0 (0%)
	CSF leak	0 (0%)
Length of stay after surgery in days (range)		4.2 (1–14)
Recurrence rate		6 (31.57%)
6 months mortality rate		7 (36.8%)
Overall mortality rate		14 (73.7%)
Post-operative complications	Respiratory	1 (5.3%)
	Deep wound	1 (5.3%)
	infection	0 (0%)
	Post op hematoma	
	Meningitis	0 (0%)
	DVT	1 (5.3%)
	PE	0 (0%)
Neurological change	No change	8 (42.1%)
	Improvement	10 (52.6%)
	deterioration	1 (5.3%)
Post vs. pre-operative ASIA	No change	14 (73.7%)
	Improvement	4 (21.1%)
	deterioration	1 (5.3%)
Radiation treatment timing	None	3 (15.8%)
	Preop	6 (31.6%)
	Postop	7 (36.8%)
	both	3 (15.8%)
Pre-operative radiation type	None	10 (52.6%)
	Fractionated	9 (47.4%)
	SRS	0 (0%)
Post op radiation type	None	9 (47.4%)
	Fractionated	2 (10.5%)
	SRS	8 (42.1%)
Discharge destination	Home	18 (94.7%)
	Rehabilitation facility	1 (5.3%)
Mean follow-up in months (range)		14.3 (1–47)

CSF = cerebrospinal fluid.

**Table 3 jcm-12-00536-t003:** Imaging and screw fixation advancements over the years in 19 patients with epidural spinal metastasis.

Years	Intraoperative Imaging		Screws			
	O-Arm	Flouroscopy	No Fixation	Sextant	Longitude	Viper-Prime
2013–2014	1 (11.1%)	8 (88.9%)	0 (0%)	1 (11.1%)	8 (88.9%)	0 (0%)
2015–2016	3 (50%)	3 (50%)	1 (16.6%)	0 (0%)	5 (83.3%)	0 (0%)
2017–2019	2 (50%)	2 (50%)	2 (50%)	0 (0%)	0 (0%)	2 (50%)
Overall	6 (31.5%)	13 (68.4%)	3 (15.7%)	1 (5.3%)	13 (68.4%)	2 (10.5%)

## Data Availability

The data presented in this study are available on request from the corresponding author.

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
