# Peer review of "The Evolution of Minimally Invasive Spine Tumor Resection and Stabilization: From K-Wires to Navigated One-Step Screws"

_jcm, 2023, doi:10.3390/jcm12020536_

Round 1

Reviewer 1 Report

1. All seems good.

2. As for an innovative surgery, I think, much more detail in description and  illustration of the method should be add.

3. In the discussion, the background of the vertebral metastasis remedy could be abbreviated, and more expanding attention could be payed in the outcome and limitation of the innovation.

Author Response

We appreciate the reviewer's comments and have revised the manuscript accordingly. We have addressed each point as detailed below. We believe that these changes have helped to clarify the content of our paper and make it worthy of publication. Thank you for considering our revised manuscript for publication. We look forward to your ultimate decision.

  1. All seems good.

Respose: Thank you for the kind review

2. As for an innovative surgery, I think, much more detail in description and  illustration of the method should be add.

Response:

Thank you for the kind review, the following sentences were added to the surgical technique:

"The internal K-wire was advanced to be 3-4mm proud relative to the screw tip."

"Once navigation confirmed the screw to be in an acceptable starting point, the internal sharp K-wire was hammered into the starting point while gradually advancing the K-wire. "

3. In the discussion, the background of the vertebral metastasis remedy could be abbreviated, and more expanding attention could be payed in the outcome and limitation of the innovation.

Response:

Thank you for the kind review, the Discussion section was changed accordingly:

"Treatment options for symptomatic patients with metastatic epidural spinal cord compression (MESCC) include a combination of corticosteroids11, radiotherapy12 or neural elements decompression through resection of compressive tumors in selected patients with or without instrumentation1,8,13. "

"The principals of conversion of open surgery to MISS includes maintaining the goals and outcomes of surgery while reducing complications rate and expediting recovery. In the current study, one-step screws were used since 2017. Screw malposition was not observed with any of the screw type and no intraoperative complications involving screw insertion were observed, thus achieving the role of MISS of obtaining the operative goals while refraining from cmplications."

"This is a retrospective study of a small cohort examining a new technology. The small numbers do not allow definite conclusions."

Reviewer 2 Report

The Evolution of Minimally Invasive Spine Tumor Resection 2 and Stabilization: from K-wires to Navigated One-step Screws

This is a well-written paper with very clear objectives. Though this is not my area of specialty, it holds a lot of interest to me considering the generally poor prognosis associated with this area of cancer management. Improvements in this area of medicine should hold a lot of interest for other readers.

This retrospective study evaluating the evolution of this surgical technique of applying minimally invasive spine tumor resection provides further evidence regarding the utility of this surgical procedure. The evidence adduced will provide information and interest for further research in this area with great potential to accelerate progress in this management technique.

The objectives of the study are clear, and the background information though not profuse is adequate for this paper. The data collection and analysis method is strong enough to support the results obtained, discussions, and conclusions of the paper.

In my opinion, though I will reiterate that this is not my area of specialty, the conclusion that the adoption of assistive devices such as intraoperative imaging and one-step screw insertion systems made this surgical procedure safe and efficient is valid regardless of the small sample size.

Author Response

We appreciate the reviewer's comments and have appreciate the positive feedback. Thank you for considering our revised manuscript for publication. We look forward to your ultimate decision.